# GC-ToF-MS Profiling and In Vitro Inhibitory Effects of Selected South African Plants against Important Mycotoxigenic Phytopathogens

**DOI:** 10.3390/life13081660

**Published:** 2023-07-30

**Authors:** Ntagi Gerald Mariri, Preachers Madimetja Dikhoba, Nkoana Ishmael Mongalo, Tshepiso Jan Makhafola

**Affiliations:** 1Center for Quality of Health and Living, Faculty of Health and Environmental Sciences, Central University of Technology, Private Bag X20539, Bloemfontein 9301, South Africa; nmariri@cut.ac.za (N.G.M.); mdikhoba@cut.ac.za (P.M.D.); 2College of Agriculture and Environmental Science (CAES), Laboratories, University of South Africa, Private BagX06, Florida 0710, South Africa

**Keywords:** mycotoxins, biofungicides, minimum inhibitory concentrations, antioxidant activity, total phenolic contents

## Abstract

The harmful effects following the ingestion of mycotoxin-contaminated food include the induction of cancers, mutagenicity, immune suppression, and toxicities that target organs of the digestive, cardiovascular, and central nervous systems. Synthetic fungicides are generally associated with a high toxic residue in food and the development of excessive fungal resistance. This study aimed to determine the antifungal activities against mycotoxigenic fungi of selected South African plant leaves and potentially develop plant-derived bio-fungicides, and, furthermore, to explore the in vitro antioxidant activity and the phytochemical spectra of the compounds of the selected medicinal plant extracts. The extracts were tested for antifungal activity against phytopathogenic strains using a microdilution broth assay. *Bauhinia galpinii* extracts exhibited the lowest minimum inhibitory concentration (MIC) against *C. cladospoides* and *P. haloterans* at 24 h incubation periods. *C. caffrum* had good antioxidant activity against 2,2-diphenyl-1-picrylhydrazyl (DPPH) with 50% inhibitory concentration (IC_50_) values of 0.013 mg/mL while *B. galpini* had IC_50_ values of 0.053 against free radicals of 2,2′-azinobis (3-ethylbenzthiazoline-6-suphonic acid (ABTS). The antimycotoxigenic and antioxidant activity exerted by both *B. galpinii* and *C. caffrum* may well be attributed to high TPC. In the GC-ToF-MS analysis, all the selected medicinal plants exhibited the presence of Hexadecanoic acid at varying % areas, while both *B. galpinii* and *C. caffum* exhibited the presence of lupeol at % area 2.99 and 3.96, respectively. The compounds identified, particularly the ones with higher % area, may well explain the biological activity observed. Although the selected medicinal plants exhibited a notable biological activity, there is a need to explore the safety profiles of these plants, both in vitro and in vivo.

## 1. Introduction

Mycotoxins are a group of secondary metabolites produced by various toxigenic strains of fungi from various fungal genera that can contaminate different agricultural commodities across the world and mostly affect developing countries, with an estimate of 25% of the world’s crop affected every year [1]. The toxic effects of mycotoxins on animal and human welfare can be mild or severe, otherwise acute or chronic, and are referred to as mycotosis. Acute toxicity may include fever, abdominal pain, portal hypertension, and death, while chronic mycotoxicosis is strongly associated with liver cirrhosis, kidney failure, immune toxicity, and cancer incidences [2]. There have been numerous mycotoxins detected to have contaminated various agricultural commodities in the world, during the different stages of harvesting including pre-harvest, and post-harvest [3]. Besides posing a threat to overall food security, such mycotoxins may well account for many devastating losses to many farmers, and may well lead to financial losses, hence the closure of farming businesses and possible hunger in many developing and underdeveloped countries. However, the groups with the greatest occurrence and toxicity may include aflatoxins, fumonisins, and ochratoxins [4]. Aflatoxins are by far amongst the most toxic mycotoxins and are produced primarily by the species belonging to the genus *Aspergillus*, specifically *Aspergillus flavus*, *Aspergillus parasiticus,* and *Aspergillus nomius* [5]. Aflatoxin contamination represents the second leading cause of hepatocellular carcinoma worldwide with an estimate of 250,000 deaths relating to hepatocellular carcinomas seen only in sub-Saharan Africa [6,7].

The widespread contamination and occurrence of fumonisins have been observed mainly in maize crops cultivated in many parts of South Africa. Fumonisins were initially reported in South Africa from moldy maize having been isolated from *Fusarium verticillioides* [6]. *Fusarium* infections are associated with neural tube defects, while significant exposure to fumonisin B1 has been suspected to enhance the risk of esophageal cancer [8]. Ochratoxins are produced by fungi belonging to the genera *Aspergillus* and *Penicillium* such as *Aspergillus alutaceus* and *Penicillium verrucossum* [9,10]. Ochratoxin A is the most toxic mycotoxin within this group and commonly contaminates food products such as maize, cocoa beans, coffee beans, cassava flour, fish, peanuts, dried fruits, wine, milk, and poultry eggs [11]. Ingestion of food contaminated by these compounds has been seen to have toxic effects on the liver and the kidneys with the ability to cause diseases of the upper urinary tract and may well induce various forms of cancers [12]. Currently, the most favored management and treatment strategy to alleviate mycotoxicosis is the application of synthetic fungicides. However, this use of chemicals is deterred by the restrictions on their use because of the high potential of toxic residues in food and, importantly, the development of fungal resistance [13]. Additionally, some of the toxic fungal strains have developed resistance to many of these synthetic fungicides [14,15]. Genetically modified crops have been relied on to reduce crop contamination. However, both this strategy and synthetic fungicides are not affordable resources to most subsistence farmers [16]. The use of medicinal plant leaves may well serve as a cheaper alternative to prevent mycotoxin infections. Although their safety is of paramount importance, plant species are consumed as medicine to treat a variety of human and animal infections and may be a safer alternative.

The aim of this study is to determine the in vitro antifungal and antioxidant activity against mycotoxigenic fungi of selected South African plants and potentially develop less expensive, easily biodegradable, and eco-friendly plant-derived bio-fungicides that can be used in place of synthetic fungicides. The phytochemical profile of such medicinal plant species is also explored. After infecting the food crops, fungal strains promote decay through oxidation. Reactive oxygen species (ROS) result in stress and are likely to cause devastating health conditions in both humans and animals when consumed.

## 2. Materials and Methods

### 2.1. Plant Selection and Collection

The leaves of the following plant species without symptoms of fungal or bacterial infections were randomly collected from the plants growing in the Lowveld Botanical Garden within Mpumalanga province of South Africa (30°57′58.16″ E 25°26′42.61″ S Long 30.96800Lat-2554669). The identity of the plants was confirmed by Mr Willem Froneman (National Biodiversity Institute, NBI) and matched with the voucher specimens as reported in Table 1 below. Only leaves were used for plant conservation considerations. 

### 2.2. Extract Preparation

The plant material was thoroughly washed, air-dried at room temperature, and ground into fine powder. Separately weighed 20 g of the powdered leaves were weighed into 500 mL Schott bottles followed by the addition of 200 mL of methanol (Analytical grade, Radchem (PTY) Ltd., Alberton, South Africa), making it 1:10 *w/v*. The Schott bottles were shaken on an orbital shaker (OrbiShake, Labotec, Seoul, Republic of Korea) at 130 RPM for three consecutive days, uninterrupted. The methanol extracts were then evaporated under reduced vacuum using a Rotary evaporator (Rotavapor R-300, Buchi, Shanghai, China) and then transferred into pre-weighed McCartney Bottles and concentrated to dryness under a stream of cold air under the fume hood for complete dryness. After drying, the McCartney Bottles were reweighed to determine the quantity of the plant material extracted. The dried plant extracts were resuspended in 20% methanol to stock solutions of 10 mg/mL to be used in subsequent bioassays [17]. 

### 2.3. Antifungal Activity

#### 2.3.1. Fungal Pathogens

A total of seven phytopathogenic fungal strains belonging to the genera *Aspergillus*, *Fusarium Penicillium,* and *Cladosporium* were selected for antifungal activity tests. The freshly prepared fungal cultures of *Aspergillus parasiticus* (PPRI: 9153), *Aspergillus nomius* (PPRI:3753), *Fusarium verticilloides* (PPRI: 10148), *Fusarium graminearum* (PPRI: 10340), *Fusarium oxysporum* (PPRI: 10185), *Penicillium haloterans* (PPRI 25804), and *Cladosporium cladospoides* (PPRI: 10367) were purchased from the Agricultural Research Council, Plant Protection Research Institute (ARC-PPRI) located at Pretoria, South Africa. Dr Adrianna Venter-Jacobs (ARC-PPRI) confirmed their identities. These fungal cultures were then sub-cultured from potato dextrose agar (PDA) slants into freshly prepared potato dextrose broth (PDB) growth medium plates.

#### 2.3.2. Inoculum Preparation

Sterile distilled water and 30% glycerol spore suspensions were prepared by gently scrubbing the conidia from the periphery of actively growing 4- to 5-day-old cultures of *Aspergillus*, *Fusarium, Penicillium*, and *Cladosporium*, respectively. Glycerol assists with coating *Aspergillus* spores and prevents them from floating in liquid broth media. Thereafter, a full loop of each spore suspension was transferred into 75 mL freshly prepared PDB and incubated overnight at 30 °C until slight turbidity is observed. Since *Fusarium* fungal strains sporulate slowly, it was necessary to obtain the suspensions by the exhaustive scraping of the surface using a sterile loop [18]. 

#### 2.3.3. Determination of Minimum Inhibitory Concentrations (MIC) and the Total Activity of Plant Extracts

The serial microplate dilution assay developed by Eloff [19], with slight modifications, was used to determine minimum inhibitory concentrations of 10 mg/mL plant extracts dissolved in 20% methanol against the selected phytopathogenic fungi. Briefly, 100 μL of the plant extracts was transferred into the first row of wells in a flat bottom sterile 96-well plate (Merck, RSA, Westfield, South Africa) laid with 100 µL of freshly prepared potato dextrose broth and the contents in the wells were serially diluted two-fold. Aliquots of 100 µL of standardized fungal cultures at a concentration of 1.1 × 10^7^ cfu/mL of each tested fungal strain were added into the corresponding wells. Wells with 20% methanol were separately prepared as negative controls while amphotericin B, Propiconazole, and Tebuconazole, obtained from Merck, South Africa, were used as standard antifungal agents (positive controls). All the wells were then loaded with 40 μL of 0.02 mg/mL of freshly prepared *p*-iodo-nitroterazolium (INT) chloride (Sigma-Aldrich, Darmstadt, Germany), as an indicator for fungal growth. The wells showing purplish color were indicative of fungal growth, whereas colorless or greenish color indicated that the plant extract inhibits fungal growth and was reported as the minimum inhibitory concentration (MIC) of the extract [20]. After the addition of INT, the plates were incubated at 30 °C under 100% humidity. The MIC results were read after 24, 48, and 72 h of incubation, respectively. The extracts were tested in triplicates. Total activity was then calculated by dividing the quantity extracted in milligrams from 1 g of plant material by the MIC value (mg/mL), to determine the total activity level of antifungal compounds of each extract [21].

### 2.4. Antioxidant Activity

#### 2.4.1. 2,2-diphenyl-1-picrylhydrazyl (DPPH) Radical Scavenging Activity

The free radical scavenging activity of the selected medicinal plant extracts was determined using the method previously described by Fadipe et al. [22], with slight modifications. The 2,2-diphenyl-1-picrylhydrazyl (DPPH) assay is based on the ability of the extract or compound to donate a hydrogen atom, thereby reducing the purplish color of DPPH to yellowish or colorless. Briefly, aliquots of plant extracts were serially diluted two-fold in methanol to yield concentrations of 0.5, 0.25, 0.13, 0.06, 0.03, 0.02, 0.01, and 0.004 mg/mL, respectively, in 100 μL of methanol inside 96-well microplates (Merck, RSA). All the wells were then loaded with 100 μL of a freshly prepared methanol solution of 0.02 mg/mL DPPH (*w/v*) (Merck, RSA) to indicate free radical scavenging activity. Then, the plates were incubated in the dark for about 1 h prior to absorbance reading at 517 nm using the microplate reader (SpectraMax iD3, Separations (Pty) Ltd., San Jose, CA, USA).

#### 2.4.2. 2,2′-azinobis (3-ethylbenzthiazoline-6-suphonic acid (ABTS) Radical Scavenging Activity

The ABTS radical scavenging activity of the selected medicinal plant extracts was measured with modifications of the 96-well plate method previously used by Re et al. [23]. The sterile 96-well microplates were laid with 100 μL of methanol followed by the two-fold serial dilution of the tested plant extracts to yield different concentrations as used in the DPPH assay above. The ABTS solution was prepared by dissolving 7 mM 2,2′-azinobis (3-ethylbenzthiazoline-6-suphonic acid (ABTS) salt and 2 mM of potassium persulphate in 3 mL distilled water and then incubated in the dark for the entire 16 h. Then, the solution was diluted 1:60 (*v/v*) with pure-grade fresh methanol. All the wells were then loaded with 100 μL of the methanol solution of ABTS. The plates were then incubated in the dark for 5 min and read at 734 nm using the microplate reader (SpectraMax iD3, Separations (Pty) Ltd.). Ascorbic acid was used as a positive control in both ABTS and DPPH assays, while wells containing methanol and a test radical were used as negative controls while the percentage of inhibition was then calculated using the formula:% inhibition = 1 − (A_t_/A_0_) × 100
where A_t_ equals the absorbance of the treated sample, and A_0_ equals the absorbance of the negative control. The concentration of the plant extract leading to a 50% reduction of DPPH (IC_50_) was then determined from the linear graphs constructed using the percentage of inhibition against the concentration of plant extract, using Microsoft Excel from three experimental replicates.

### 2.5. Phytochemical Screening

#### 2.5.1. Determination of Total Phenolics

Total phenolics were determined using the Folin–Ciocalteu method as previously described by Adebayo et al. [24]. In short, 25 μL of the crude extracts dissolved in dimethyl sulphoxide (DMSO) were treated with 250 μL of Folin–Ciocalteu reagent for 5 min. The reaction was stopped by adding 750 μL 20% anhydrous sodium carbonate and the volume was made up to 5 mL with ultrapure distilled water and incubated in the dark at room temperature for 2 h. The absorbance was then read at 760 nm using the microplate reader (SpectraMax iD3, Separations (Pty) Ltd.). The phenolic contents of the selected medicinal plants were then determined from a standard curve of different concentrations of gallic acid dissolved in DMSO and the results were expressed as mg/g gallic acid equivalent (GAE).

#### 2.5.2. Determination of Flavonoids

The flavonoid contents of the tested plant extracts were determined according to the method described previously by Adebayo et al. [24], with slight modifications. Shortly, 100 μL of each crude extract was dissolved in 300 μL of methanol, to which 20 μL of 10% aluminium chloride was added. An additional 20 μL of 1 M sodium acetate was added to the solution. The resulting solution was then made up to 1 mL with ultrapure distilled water and incubated at room temperature for 30 min in a microplate. About 10 mM of quercetin was used as a standard. The absorbance was read at 450 nm in a microplate reader (SpectraMax iD3, Separations Pty Ltd.). The flavonoid contents of each extract were then expressed as mg/g quercetin equivalent (QE).

#### 2.5.3. Phytochemical Analysis of Crude Extracts by Gas-Chromatography Time of Flight (GC-ToF-MS) Mass Spectrometry

The GC-ToF-MS analysis of organic extracts of the selected medicinal plants species was carried out using method adopted from Mongalo and Raletsena [25], with slight modification. The little amount of about 0.05 mg/mL of the selected medicinal plants extracts were completely dissolved in acetonitrile (GC-MS grade, Sigma Aldrich, Germany) to the lowest concentration. The separation of compounds was performed on a gas chromatography (7890 N GC-ToF-MS, Agilent Technologies, Santa Clara, CA, USA) coupled to a LECO Pegasus HT Flight Mass Spectrometry Time (ToF-MS) obtained from LECO Corporation, Michigan, USA. In short, the prepared samples were loaded into a Gerstel MPS2 Liquid/HS/SPME auto-sampler. For the chromatographic separation, AJ & W capillary column HP-5MS 30 × 0.25 mm I.D with a film thickness of about 0.25 µM was used.

The following conditions were applied in the chromatographic separation: About 1 µL of the sample was injected at 250 °C with a spitless injector. The GC oven was programmed at 80 °C for 1 min, then ramped up at 10 °C per min to 280 °C for 20 min. Helium obtained from Afrox (Johannesburg, RSA) at 99.99% purity was used as a carrier at a constant flow of 1 mL/min. The interface temperature of the GC-ToF-MS was set at 280 °C and the mass spectra were obtained in full scan mode at 70 ev (*m/z* scan varying from 50 to 550). The collection of data was obtained using ChromaToF, which possesses a NIST 95 library for compound matches.

## 3. Results

### 3.1. Percentage Yields, and Antifungal and Total Activity

The percentage yield for each of the tested plant extracts was determined by dividing the dried extracted mass after the evaporation of the solvent by the dried plant mass used for extraction multiplied by 100 [21]. *Combretum caffrum* yielded 23% of crude extract and was the highest percentage yield recorded from all the tested plant species in this study, followed by *Bauhina galpinii* which yielded 21.4% of the crude extract, while the lowest percentage yield was recorded from *Markhamia obtusifolia* at 10.7% (Table 2). The antimicrobial activity of the extracts from the selected medicinal plants are reported in Table 2. The methanol extracts from *Bauhinia galpinii* leaves exhibited the notably lowest minimum inhibitory concentration (MIC) value of 0.16 mg/mL against *Furasium. graminearum* at a 24, 48, and 72 h incubation period, and a similar MIC against *Cladosporium cladosporioides* at the 24 and 48 h incubation time. The extract further yielded an MIC of 0.31 mg/mL against both *Aspergillus parasiticus* and *Asparagus nomium* at all three different incubation periods. The extract from *Combretum caffrum* exhibited an MIC value of 0.31 mg/mL against *Furasium verticilloides, F. graminareum,* and *C. cladosporioides* at all tested incubation periods, and the notably lowest MIC value of 0.16 mg/mL against *C. cladospoides* at both the 24 and 48 h incubation period. *Markhamia obtusifolia* extracts exhibited an MIC value of 0.31 against *A. nomius* and *F. verticilloides* at all tested time intervals while *Maytenus undata* exhibited an MIC value of 1.25 mg/mL against both *F. graminareum* and *Aspergillus parasiticus* at 48 and 72 hr incubation. All the selected strains were more susceptible to Propiconazole, Amphotericin B, and Tebuconazole as antifungal agents.

Total activities (TAs) of the selected medicinal plants are presented in Table 3. The extracts from *Combretum caffrum* exhibited the highest total activity yielding a TA value of 1437.5 mg/mL against *Aspergillus nomius* at the 24 and 48 h incubation period while *Bauhinia galpinii* extracts had a total activity of 1337.5 mL/g against *C. cladospoides* at the 24 and 48 h incubation times and a similar TA against *Penicillium haloterans* at the 24 h incubation period. *Maytenus undata* extracts exhibited the lowest TA values yielding TA value of 139.2 against all three Furasium (*F. verticilloides*, *F. graminearum,* and *F. oxysporum*) strains and *A. parasiticicus* at various incubation periods while *M. obtusifolia* exhibited the lowest TA value of 169.84 against both *P. haloterans* and *C. cladospoides* at the 72 h incubation period.

### 3.2. Antioxidant Activity

The free radical scavenging activity of the selected medicinal plants against 2,2-diphenyl-1-picrylhydrazyl (DPPH) and 2,2′-azinobis (3-ethylbenzthiazoline-6-suphonic acid (ABTS) assays are presented as an inhibitory concentration leading to a 50% reduction of free radicals (IC_50_) as summarized (Table 4). The extracts from *Combretum caffrum* had the lowest antioxidant activity against DPPH with IC_50_ values of 10 and 70 µg/mL against DPPH and ABTS, respectively, while *Bauhina galpinii* had a notable IC_50_ value of 50 µg/mL against the ABTS free radical. *Maytenus undata* extracts yielded an IC_50_ value of 30 µg/mL against the DPPH free radical and a further 70 µg/mL against ABTS while *Markhamia obtusifolia* exerted a notable antioxidant activity against DPPH, yielding an IC_50_ value of 60 µg/mL. Ascorbic acid exhibited IC_50_ values of 5.7 and 4.0 µg/mL against DPPH and ABTS free radicals, respectively. The highest ABTS/DPPH value was obtained from the *B. galpinii* extract yielding 9.8, while *M. obtusifolia* and *M. undata* exhibited 2.83 and 2.33, respectively.

### 3.3. Phytochemical Analysis of the Selected Medicinal Plants

#### 3.3.1. Total Phenolic and Flavonoid Contents

The total phenolic contents (TPCs) and total flavonoid contents (TFCs) of the selected medicinal plants are presented below (Table 5). *Combretum caffrum* and *Bauhinia galpinii* extracts exhibited higher TPC values yielding 0.46 and 0.41 mg/g GAE, respectively, while extracts from *Maytenus undata* and *Markhamia obtusifolia* exhibited significantly higher TFC values of 0.39 and 0.13 (*p*-values ≤ 0.05), respectively.

#### 3.3.2. Gas Chromatography Time-of Flight Mass-Spectrometry (GC-ToF-MS) Profiling of Selected South African Plants

*Bauhinia galpinii* revealed a total of thirty-six compounds, twelve compounds of which were present in abundance, identified by a % area greater than one. The chromatograms of all the medicinal plants extracts are presented as a Appendix A. The compounds from *B. galpinii* with their retention time and relative abundance (% area) in the crude extract are presented in Table 6.

Compounds such as 4-1,1-dimethylpropyl (19.23%), hexadecanoic acid (9.07%), and Stigmast-5-en-3-ol, 3. beta.,24S (3.43%) are amongst compounds that had a higher % area. Hexadecanoic acid exhibited the highest % identification match (98%), followed by Stigmast-5-en-3-ol, (3. beta.,24S)—(93%) and Phenol, 4-(1,1-dimethylpropyl)—(92%), while both lupeol and n-Tricosane exhibited a 90% identification match.

Fourteen compounds were found to be abundant in the *Combretum caffrum* crude organic extracts, while the compounds with their retention time and relative abundance (% area) in the crude extract are presented in Table 7. Compounds such as 4-(1,1-dimethylpropyl (which yielded 22%) and hexadecanoic acid (7.28%) were the most abundant compounds identified in the organic crude extracts from *C. caffrum*. Furthermore, compounds such as Croweacin, Hexadecanoic acid, 8-Octadecenoic acid, Eicosanoic acid, Octadecanoic acid, Stigmast-5-en-3-ol, and Docosanoic acid exhibited the highest identification match of 99%.

From the analysis of the extracts from *M. obstufolia,* a total of seventy-seven compounds were identified and nine of these compounds were identified to have a % area greater than one and, therefore, are concluded to be present in abundance. Table 8 presents the compounds with their retention time and relative abundance (% area) in the crude extract. The extract had phenol, 4-(1,1-dimethylpropyl)—(20.70%) and hexadecanoic acid (12.05%) as the most abundant compounds; 9-Octadecenoic acid (Z)—was also identified with a % area of 6.97, which indicated its abundance.

*Maytenus undata* revealed a total of eight compounds that were present in abundance. The compounds with their retention time and relative abundance (% area) in the crude extract are presented in Table 9. The compound with the highest peak area percentage was phenol, 4-(1,1-dimethylpropyl)- accounting for 16.82%. Other compounds that dominated the chromatogram of the organic crude extracts were gamma-Sitosterol (Clionasterol) (4.29%) and hexadecanoic acid (3.82%).

## 4. Discussion

Numerous mycotoxigenic fungal strains negatively impact the production of food crops consumed by humans as food. Such strains may well reduce both the crop yields (quantity) and general composition (quality) of foods and contribute to overall hunger, compromised food security, and devastating illnesses when consumed [26,27]. The situation is further compounded by the projection that the total global food demand is expected to increase by 35% to 56% between 2010 and 2050, hence observed in many African countries [28,29]. The use of medicinal plants as cheap and readily available possible pesticides to counter various fungal infections on crops may well be a possible solution to counter food infections, thereby increasing both yields and quality, and thereby alleviating hunger in both developing and underdeveloped countries [30,31,32,33]. The antifungal activity of some selected medicinal plants is depicted in Table 2. *Bauhinia galpinii* methanol extracts exhibited a notable minimum inhibitory concentration (MIC) compared to other medicinal plants. The extract yielded the lowest MIC value of 0.16 mg/mL against a plethora of pathogens that includes *Furasium graminareum*, *Furasium oxysporum*, *Penicillium haloterans,* and *Cladosporum cladospoides* at a 24 h incubation period. As there are no validated endpoint criteria for in vitro testing of plant extracts [34,35], Souza et al. [36] proposed that extracts with an MIC < 0.5 mg/mL were considered strong inhibitors while those with an MIC between 0.5 and 1.6 mg/mL were considered moderate inhibitors, and extracts with MIC > 1.6 mg/mL were considered weak inhibitors for at least two different temperature regimes [37]. Using the standard above, methanol extracts from *Bauhinia galpinii* remain the strongest inhibitor of *F. graminareaum* and *C. cladospoides,* yielding an MIC value of 0.16 mg/mL at 24 and 48 h. Recently, *F. graminareum* was reported to cause *Fusarium* head blight (FHB) on wheat, barley, and other grains by secreting hundreds of unknown putative effectors which may well favor the progression of the disease through interference with crop immunity, thereby promoting the formation of higher quantities of mycotoxins [38,39]. Although the extracts had a notable inhibition of the fungal strain, there is a need to further quantify the mycotoxin production in an in vivo experimental setup. Furthermore, the mode of action needs to be explored. According to other studies, the mode of action is believed to be through mycelial growth inhibition [12]. Elsewhere, the acetone extracts from *B. galpinii* leaves exhibited a noteworthy antifungal activity against a plethora of both human- and crop-infecting fungal strains [40,41,42]. However, these results are not comparable with the findings of the current study due to differences in terms of the nature and origin of the strains, the collection site of the plant specimen, and several other environmental conditions.

The methanol extracts from *Combretum caffrum*, *Markhamia obtusifolia,* and *Maytenus undata* also exhibited a strong inhibition against at least one fungal strain for at two different temperatures. *C. caffrum* exhibited an MIC value of 0.31 mg/mL against two *Furasium* strains such as *F. vercitilloides* and *F. graminearum*. According to reports, the extracts with a similar MIC at two different temperatures might mean that the extracts are fungistatic and, if used in a ploughing field to encounter mycotoxin infection, should be administered frequently until the fungal strains die off [43,44]. *F. verticilloides* mainly infect maize and are notoriously known to produce harmful secondary metabolites known as fumosinin mycotoxin which causes severe human and animal diseases [45,46]. Elsewhere, the acetone extracts from *C. caffrum* exhibited a potent antifungal activity, both in vitro and in vivo, against a plethora of both human- and crop-infecting pathogens, and hence possess other biological activities [47,48,49]. Earlier, the compounds from *C. caffrum* leaves such as combrestatin A-4 and Combrestatin A-4P and some derivatives thereof were found to be highly active against a plethora of pathogens such as *Fusarium oxysporum*, *Sclerotinia sclerotiorum*, *Pyricularia oryzae,* and *Rhizoctonia solani* [50].

The methanol extracts from the selected medicinal plants exhibited some varying degrees in the total activity, which serves as a guide to the concentration at which the plant extract could be diluted and still inhibit the fungal growth (Table 3). *Combretum caffrum* exhibited the highest total activity (TA) of 1437.5 mL/g. This means the extract (1g) could be diluted with such a high volume of water and still inhibit the growth of the fungal pathogen, *Aspergillus nomius* [22,51]. It is of paramount importance to note that the TA is dependent on several factors which includes the solubility of the plant materials in a specific solvent and the activity of such an extract against the selected micro-organisms [12]. The results in the current work may well explain that these plant extracts can well mitigate the risk of mycotoxin contamination. Although the extracts exhibited a notable antimycotoxigenic activity, there is still a need to explore the minimum bactericidal concentrations, to determine and further explore whether the extracts are fungicidal or fungistatic. Furthermore, there is a need to explore the safety profiles and mode of action, and quantify such mycotoxins using new technology and recent analytical tools.

Reactive oxygen species (ROS) may well arise in edible crops consumed as food on a daily basis as a by-product of several metabolic processes that are in different cell compartments, or because of the inevitable escape of electrons to oxygen from the electron transport activities of chloroplasts, mitochondria, and plasma membranes [52,53]. It is important to note that these reactive species are formed in chloroplasts, mitochondria, plasma membranes, peroxisomes, apoplasts, the endoplasmic reticulum, and cell walls. The free radical scavenging activity of the methanol extracts from the selected medicinal plants are depicted in Table 4, against 2,2-diphenyl-2-picrylhydrazyl (DPPH) and 2,2′-azinobis (3-ethylbenzthiazoline-6-suphonic acid (ABTS) free radicals. The DPPH assay is common and widely used in the antioxidant capacity screening of plant extracts, isolated compounds, and their respective derivatives, fruits and vegetables juices or extracts thereof, for it is easy and rapid, and requires only a UV-vis spectrophotometer to test [54]. Compared with the ABTS assay, the DPPH radical is commercially available and does not have to be generated before the assay; ABTS•^+^
*Combretum caffrum* exhibited a noteworthy and the lowest 50% inhibition (IC_50_) of DPPH yielding an IC_50_ value of 10 µg/mL, while *Markhamia obtusifolia* exhibited an IC_50_ value of 70 µg/mL against the ABTS free radical. According to More and Makola [55], medicinal plant extracts regarded as potent in an antioxidant assay should yield the lowest IC_50_ value of ≤20 µg/mL. Other authors corroborate the similar benchmark [56,57], while other have reported an IC_50_ as high as over 200 µg/mL as in the case with *B. galpinii* in the current work [58,59]. Elsewhere, the fractions from the 70% acetone extracts of *B. galpinii* leaves exhibited DPPH inhibition, yielding an IC_50_ as low as 0.89 and 4.30 µg/mL [40]. Besides speculating that the antioxidant compounds might be much more soluble in 70% acetone, the results could not well be compared to our current study due to differences in terms of the extracting solvent and several other parameters. Judging by the ABTS/DPPH correlations (ADC), *M. undata* exhibited a notable ADC correlation of 2.33, which is over three-fold that of ADC od ascorbic acid which was used as a control drug. From these findings, except in the case of *B. galpinii*, the extracts exhibited higher inhibition compared to ABTS, contrary to other authors [53].

Phenolics embedded within medicinal plants are sought after more often as inhibitors of oxidative stress, hence accounting for various biological activities of such plants []. In the phenolic content analysis (Table 5), three of the four medicinal plants reported a higher phenolic content (TPC), compared to their total flavonoid compounds. A similar trend has been reported elsewhere [60,61,62]. These higher TPC contents may well explain the antioxidative ability of the medicinal plants selected for the current study. In the GC-ToF-MS analysis, the plant extracts yielded varying compounds at varying time intervals and identification matches (Table 6, Table 7, Table 8 and Table 9). Comparing the spectra of the selected medicinal plants, all the extracts contained Hexadecanoic acid (HA) and methyl ester. Recently, HA was identified from a variety of medicinal plants and was reported to possess an enormous antimicrobial and antioxidant activity [63,64,65,66]. Both *B. galpinii* and *C. caffum* exhibited the presence of lupeol at a % area of 2.99 and 3.96, respectively, while *M. obtusifolia* and *M. undata* contained phytol at a % area of 2.80 and 1.78, respectively. Besides its known antioxidant activity, Lupeol and its derivatives were reported to possess enormous anticancer, wound-healing, and anti-inflammatory activities [67,68,69]. Elsewhere, phytol was found to possess a notable immunostumulant and antimicrobial activity [70].

## Figures and Tables

**Table 1 life-13-01660-t001:** Plant species studied and their voucher specimen numbers.

Plant Species	Family	Voucher Number
*Bauhinia galpinii* N. E. Br.	Fabaceae	Glow 27/1986
*Combretum caffrum* (Eckl. & Zeyh.) Kuntze	Combretaceae	Glow 92/1997
*Markhamia obstufolia* (Baker) Sprague	Bignoniaceae	Glow 16/1994
*Maytenus undata* (Thumb.) Blakelock	Celastraceae	Glow 157/1986

**Table 2 life-13-01660-t002:** Minimum inhibitory concentrations (mg/mL) of the selected methanol leaf extracts against the tested fungi recorded after 24, 48, and 72 h incubation.

Plant Species	Extraction	*F. verticilloides*			*F. graminearum*			*F. oxysporum*			*A. parasiticus*			*A. nomius*			*P. haloterans*			*C. cladospoides*		
	Yield (%)	24 h	48 h	72 h	24 h	48 h	72 h	24 h	48 h	72 h	24 h	48 h	72 h	24 h	48 h	72 h	24 h	48 h	72 h	24 h	48 h	72 h
*Bauhinia galpinii*	21.4	0.63	0.63	**0.31**	**0.16**	**0.16**	**0.16**	**0.16**	1.25	1.25	**0.31**	**0.31**	**0.31**	**0.31**	**0.31**	**0.31**	**0.16**	**0.31**	0.63	**0.16**	**0.16**	**0.31**
*Combretum caffrum*	23	**0.31**	**0.31**	**0.31**	**0.31**	**0.31**	**0.31**	**0.31**	0.63	0.63	0.63	1.25	1.25	**0.16**	**0.16**	**0.31**	**0.31**	0.63	0.63	**0.31**	**0.31**	**0.31**
*Markhamia obtusifolia*	10.7	**0.31**	**0.31**	**0.31**	0.63	0.63	**0.31**	**0.31**	0.63	1.25	**0.31**	**0.31**	**0.31**	**0.31**	**0.31**	**0.31**	**0.31**	**0.31**	0.63	**0.31**	**0.31**	0.63
*Maytenus undata*	17.4	0.63	1.25	1.25	1.25	1.25	**0.31**	**0.31**	1.25	1.25	1.25	1.25	1.25	0.63	0.63	0.63	**0.31**	**0.63**	1.25	**0.31**	**0.31**	0.63
Amphotericin B	-	0.16	0.16	0.16	0.02	0.04	0.08	0.08	0.16	0.16	0.16	0.16	0.31	0.04	0.04	0.04	0.08	0.16	0.63	0.16	0.16	0.31
Propiconazole	-	0.08	0.16	0.16	0.04	0.04	0.02	0.02	0.02	0.02	0.08	0.16	0.16	0.02	0.02	0.04	0.02	0.02	0.02	0.02	0.02	0.02
Tebuconazole	-	0.08	0.16	0.16	0.04	0.04	0.04	0.04	0.04	0.04	0.08	0.16	0.16	0.02	0.02	0.04	0.04	0.08	0.31	0.02	0.02	0.04

Bold-faceted data show noteworthy antimicrobial activity.

**Table 3 life-13-01660-t003:** Total activity (mL/g) of the selected methanol leaf extracts against the tested fungi recorded after 24, 48, and 72 h incubation.

Plant Species	*F. verticilloides*			*F. graminearum*			*F. oxysporum*			*A. parasiticus*			*A. nomius*			*P. haloterans*			*C. cladospoides*		
	24 h	48 h	72 h	24 h	48 h	24 h	48 h	24 h	48 h	24 h	48 h	72 h	24 h	48 h	72 h	24 h	48 h	72 h	24 h	48 h	72 h
*B. galpinii*	339.68	339.68	690.32	690.32	690.32	690.32	690.32	690.32	690.32	690.32	690.32	690.32	690.32	690.32	690.32	**1337.5**	690.32	339.68	**1337.5**	**1337.5**	690.32
*C. caffrum*	741.94	741.94	741.94	365.08	184	365.08	184	365.08	184	365.08	184	184	**1437.5**	**1437.5**	741.94	741.94	365.08	365.08	741.94	741.94	741.94
*M. obtusifolia*	345.16	345.16	345.16	345.16	345.16	345.16	345.16	345.16	345.16	345.16	345.16	345.16	345.16	345.16	345.16	345.16	345.16	169.84	345.16	345.16	169.84
*M. undata*	276.19	139.2	139.2	139.2	139.2	139.2	139.2	139.2	139.2	139.2	139.2	139.2	276.19	276.19	276.19	561.29	276.19	139.2	561.29	561.29	276.19

Bold-faceted data show noteworthy total activity.

**Table 4 life-13-01660-t004:** Antioxidant activity (IC_50_ in µg/mL) of selected medicinal plants.

Plant Species	DPPH	ABTS	ABTS/DPPH
*B. galpinii*	490 ± 3.11	**50** **± 0.01**	9.8 ± 0.001
*C. caffrum*	**10** **± 0.06**	**70** **± 0.23**	7.0 ± 0.002
*M. obtusifolia*	**60** **± 0.08**	170 ± 2.24	**2.83** **± 0.01**
*M. undata*	**30** **± 0.01**	**70**	**2.33** **± 0.01**
**Ascorbic acid**	**5.7** **± 0.001**	**4.0** **± 0.01**	**0.70** **± 0.01**

Bold-faceted data show noteworthy antioxidant activity/correlation. Results recorded as mean ± SEM.

**Table 5 life-13-01660-t005:** Total phenolic and total flavonoid contents of the selected medicinal plants.

Plant Species	Total Phenolic Content (mg/g GAE)	Total Flavonoid Content (mg/g QE)
*B. galpinii*	**0** **.** **41 ± 0** **.11**	0.11 ± 0.01
*C. caffrum*	**0** **.** **46 ± 0** **.09**	0.09 ± 0.01
*M. obtusifolia*	0.16 ± 0.04	**0** **.** **13 ± 0** **.01**
*M. undata*	0.29 ± 0.12	**0** **.** **39 ± 0** **.04**

Bold-faceted data show noteworthy TPC and TFC. Results recorded as mean ± SEM.

**Table 6 life-13-01660-t006:** GC-ToF-MS analysis of *Bauhinia galpinii* crude organic extract (% area > 1).

Retention Time	Area (%)	Library Identification	Identification Match Quality (%)
**6** **.** **03**	1.98	Nonane, 4,5-dimethyl-	55
**6** **.** **17**	**3.10**	**2-Decene, 7-methyl-, (Z)-**	**46**
**6** **.** **32**	**2.80**	**Hexane, 2,3,4-trimethyl-**	**50**
**7.58**	**19.23**	**Phenol, 4-(1,1-dimethylpropyl)-**	**92**
**8.63**	1.46	Tetracosane	80
**8.76**	1.57	Heptadecane	72
**8** **.** **92**	1.25	Octacosane	64
**9** **.** **54**	1.13	2-Quinolinecarboxaldehyde, 8-hydroxy	20
**9** **.** **60**	1.47	2-Octene, 4-ethyl	38
**9** **.** **75**	1.93	1-Undecene, 7-methyl	38
**9** **.** **86**	**2.15**	**Tetracosane**	**64**
**10** **.** **02**	1.48	Cyclopentane, (2-methylbutyl)-	46
**11** **.** **91**	**7.76**	**6-Phenylisoquinoline**	**64**
**13** **.** **09**	**2.34**	**Cyclohexane, 1,2,4-trimethyl-**	**50**
**13** **.** **21**	1.87	Decanedioic acid, didecyl ester	49
**13** **.** **54**	1.06	Cyclohexane, 1-ethyl-2-propyl-	30
**15** **.** **08**	1.35	Octacosane	86
**15** **.** **13**	1.95	**Hexadecanoic acid**	**98**
**15** **.** **63**	**9.07**	Hexadecanoic acid	98
**15** **.** **76**	1.19	Palmitic Acid	50
**17** **.** **21**	1.20	Oxirane, 2-butyl-3-methyl-	46
**17** **.** **35**	1.47	2-Hexadecen-1-ol, 3,7,11,15-tetramethyl-, [R- [R*, R*-(E)]]-	64
**19** **.** **38**	**4.53**	**n-Tricosane**	**90**
**20** **.** **39**	**2.23**	**n-Tetracosane**	**86**
**20** **.** **88**	**8.63**	**Sclerodione**	**83**
**22** **.** **33**	1.18	n-Hexacosane	94
**26** **.** **05**	1.48	3,11-Dimethyl-nonacosane	27
**28** **.** **37**	**3.43**	**Stigmast-5-en-3-ol, (3. beta.,24S)-**	**93**
**28** **.** **78**	1.73	Lupenone	64
**28** **.** **96**	**2.99**	**Lupeol (Fagarasterol)(beta-Viscol)**	**90**

Bold-faceted data show compounds with high % area.

**Table 7 life-13-01660-t007:** GC-ToF-MS analysis of *Combretum caffrum* crude organic extracts (% area > 1).

**Retention Time**	**Area (%)**	**Library Identification**	**Identification Match Quality (%)**
**6.17**	1.11	2-Decene, 7-methyl-, (Z)-	52
**7.59**	**22.00**	**Phenol, 4-(1,1-dimethylpropyl)-**	**92**
**9.60**	1.36	Croweacin	99
**11.91**	**7.23**	**Phenol, 2,4-bis(1,1-dimethylpropyl)-**	**91**
**15.14**	**4.98**	**Hexadecanoic acid, methyl ester**	**99**
**15.66**	**7.28**	**Hexadecanoic acid**	**99**
**17.21**	**2.15**	**8-Octadecenoic acid, methyl ester**	**99**
**17.35**	1.75	2-Hexadecen-1-ol, 3,7,11,15-tetramethyl-, [R-[R*, R*-(E)]]-	91
**17.50**	1.55	Octadecanoic acid, methyl ester	99
**19.38**	1.57	n-Tricosane	83
**19.68**	**3.22**	**Eicosanoic acid, methyl ester**	**99**
**20.88**	**2.55**	**Sclerodione**	**83**
**21.68**	1.15	Docosanoic acid, methyl ester	99
**21.91**	**2.58**	**2,2’-Dimethyl-4,4’,5,5’-tetramethoxybiphenyl**	**90**
**22.22**	1.30	3,8,9-Trimethoxy-6H-dibenzo[b,d]pyran-6-one	80
**24.64**	**2.05**	**Dimethyl 4,6-dioxo-5,6-dihydro-4H-pyrido [3,2,1-jk] carbazole-5-spirocyclohexane-1,3-dicarboxylate**	**42**
**24.83**	**2.06**	**Dimethyl 4,6-dioxo-5,6-dihydro-4H-pyrido [3,2,1-jk] carbazole-5-spirocyclohexane-1,3-dicarboxylate**	**59**
**27.92**	1.03	Stigmasterol	91
**28.37**	**2.79**	**Stigmast-5-en-3-ol, (3. beta.,24S)-**	**99**
**28.63**	**2.21**	**Viminalol**	**93**
**28.80**	**2.45**	**Lupenone**	**55**
**28.97**	**3.96**	**Lupeol (Fagarasterol)(beta-Viscol)**	**83**
**29.95**	1.31	D: A-Friedooleanan-3-one	70

Bold-faceted data show compounds with high % area.

**Table 8 life-13-01660-t008:** GC-ToF-MS analysis of *Markhamia obstufolia* crude organic extracts (% area > 1).

Retention Time (mins)	Area (%)	Library Identification	Identification Match Quality (%)
**7** **.** **63**	**20** **.** **70**	**Phenol, 4-(1,1-dimethylpropyl)-**	**97**
**9** **.** **61**	1.11	Croweacin	98
**11** **.** **93**	**7** **.** **41**	**Phenol, 2,4-bis(1,1-dimethylpropyl)-**	**64**
**13** **.** **18**	1.92	(-)-Loliolide	91
**15** **.** **08**	1.30	9-Hexadecenoic acid, methyl ester, (Z)-	90
**15** **.** **14**	**3** **.** **02**	**Hexadecanoic acid, methyl ester**	**99**
**15** **.** **79**	**12** **.** **05**	**Hexadecanoic acid**	**99**
**17** **.** **22**	**2** **.** **51**	**8-Octadecenoic acid, methyl ester**	**99**
**17** **.** **36**	**2** **.** **80**	**Phytol**	**78**
**17** **.** **77**	**6** **.** **97**	**9-Octadecenoic acid (Z)-**	**97**
**18** **.** **00**	**3** **.** **24**	**Octadecanoic acid**	**98**
**19** **.** **39**	1.36	n-Tricosane	83
**20** **.** **89**	1.29	(+/−)-scleroderodione	83
**23** **.** **25**	1.07	n-Heptacosane	91
**25** **.** **00**	1.24	n-Nonacosane	93
**28** **.** **38**	**4** **.** **06**	**gamma-Sitosterol (Clionasterol)**	**99**

Bold-faceted data show compounds with high % area.

**Table 9 life-13-01660-t009:** GC-ToF-MS analysis of *Maytenus undata* crude organic extracts (% area > 1).

Retention Time (mins)	Area (%)	Library Identification	Identification Match Quality (%)
**7.62**	**16.82**	**Phenol, 4-(1,1-dimethylpropyl)-**	**97**
**11.93**	**7.89**	**p-Hydroxymephenytoin**	**64**
**14.04**	1.04	Neophytadiene	91
**15.70**	**3.82**	**Hexadecanoic acid**	**99**
**17.36**	1.74	Phytol	87
**24.41**	1.59	2,6,10,14,18,22-Tetracosahexaene, 2,6,10,15,19,23-hexamethyl-	98
**27.93**	1.45	Stigmasterol	91
**28.39**	**4.29**	**gamma-Sitosterol (Clionasterol)**	**99**
**28.69**	1.43	beta. -Ionone	46
**28** **.** **85**	1.01	Lupenone	30
**28.98**	1.92	Lup-20(29)-en-3-ol, (3. beta.)-	90
**29.06**	1.54	Simiarenol	86
**29.47**	1.07	Spirohexane-5-carboxylic acid, 1,1,2,2-tetramethyl-, methyl ester	25
**29.55**	**3.34**	**Fern-7-en-3beta-ol**	**93**
**29.71**	1.09	Lup-20(29)-en-3α-ol, acetate	46
**29.80**	**4.02**	**Longifolenaldehyde**	**53**
**29.97**	**10.30**	**Friedelan-3-one (Friedelin)**	**45**
**30.01**	**6.84**	**Phytol acetate**	**91**

Bold-faceted data show compounds with high % area.

## Data Availability

The data supporting the article are kept with the authors and are available upon request.

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
