# Peer review of "GC-ToF-MS Profiling and In Vitro Inhibitory Effects of Selected South African Plants against Important Mycotoxigenic Phytopathogens"

_life, 2023, doi:10.3390/life13081660_

Round 1

Reviewer 1 Report

The manuscript accounts of a study of the inhibitory effect of methanol extracts of leaves of four South African plants against several mycotoxicogenic pathogens and GC-ToF-MS profiling of the extracts. This paper may be of value due to the importance of the problem of mycotoxin poisoning.

The manuscript is rather untidely written and requires more careful editing.

Title: The title should be optimized as “GC-ToF-MS profiling […] against the important mycotoxi-3 genic phytopathogens” does not make sense

Page 1, Lines 13/14: “associated with a high for toxic residues in…” something missing in this statement

Lines 18/19: “Furthermore, to explore the in vitro antioxidant activity and the phytochemical spectra of the compounds that could well explain such biological activities.”, incomplete sentence

Line 29: “0,013”, please change to “0.013”

Kine 30: “0,053”, please change “0.053”

P.8, lines 3-4: Please give species names in italics

Table 4: Please provide also SD or SEM values

Table 5: “0,41”, please change to “0.41” and so on

Tables 6-9. Again, please use “.” As a decimal separator. If the analysis was performed more than once, please provide SD or SEM

Page 13, line 156: “B. galpinii”, please in italics

Line 167: “[].”, please include citation

Author Response

Good evening.

The authors have addressed all the comments as required by the Reviewer. Please see the attachment.

Thank you.

Reviewer 2 Report

Overall, the manuscript is well prepared. I have only minor comments:

Please shorten the abstract, it is too long.

Please extend the introduction.

Add for all devices the city and country, i.e. in line 165.

Pleas write in all methods if you analyzed dry or fresh samples.
